# A potential relationship between soil disinfestation efficacy and leaf green reflectance

Steven B. Kim[1]☯, Steven A. Fennimore[2]☯, Dong Sub Kim [2,3]*

**1** Department of Mathematics and Statistics, California State University, Monterey Bay, Seaside, CA, United States of America, **2** Department of Plant Sciences, University of California, Davis, Salinas, CA, United States of America, **3** Department of Horticulture, Kongju National University, Yesan, South Korea

☯ These authors contributed equally to this work.
* dongsub@kongju.ac.kr

**Data Availability Statement:** All relevant data are within the paper and its Supporting Information files.

**Funding:** This work was supported by the National Institute of Food and Agriculture USDA-NIFA

## Abstract

Soil disinfestation with steam was evaluated as an alternative to fumigation. Following soil disinfestation, plant health has traditionally been measured using plant size and yield. Plant health can be measured in a timely manner more efficiently, more easily and non-destructively using image analysis. We hypothesized that plant health could be quantified and treatments can be differentiated using an RGB (Red, Green, Blue) image analysis program, particularly by observing the greenness of plant leaves. However, plant size or the proportion of green area could be unreliable due to plant loss and camera's position and angle. To this end, we decided to evaluate plant health by analyzing the RGB codes associated with the green color only, which detects the chlorophyll reflectance and nutrient status, noting that the degree of greenness within the green-leaf-area was not affected by the plant size. We identified five RGB codes that are commonly observed in the plant leaves and ordered them from dark green to light green. Among the five RGB codes, the relative percentage covered by the darkest green to the lightest green was significantly different between the steam and chloropicrin treatments and the control, and it was not significantly different between the steam and chloropicrin treatments. Furthermore, the result was correlated with the total yield, and the trend observed in the first year was replicated in the second year of this experiment. In this study, we demonstrate that the RGB image analysis can be used as an early marker of the treatment effect on the plant health and productivity.

## Introduction

Methyl bromide (MB) is a soil fumigant that has been widely used for soil disinfestation prior to planting strawberry. However, the use of MB has mostly been eliminated due to restrictions mandated by the Montreal Protocol [1]. MB alternative fumigants are chloropicrin, metam sodium and 1,3-D. These fumigants are highly regulated and difficult to use in many fields due to the proximity of schools, hospitals and other sensitive sites [2]. Therefore, there is need for non-fumigant alternatives. Steam is among the non-fumigant alternatives that have been

Methyl Bromide Transitions Program in the form of a grant (2017-04265) to DSK and SAF. The funders had no role in study design, data collection and analysis, decision to publish, or preparation of the manuscript.

**Competing interests:** The authors have declared that no competing interests exist.

evaluated [2–4]. Samtani et al. [5] reported that the steam treatment is as effective as the MB treatment for reducing soil pests such as phytopathogenic fungi, bacteria, nematodes, and weeds. Other studies also support that the steam controls major pathogens, weeds, and nematodes in strawberry fields [4, 6, 7].

The efficacy of steam on soil pests has been typically evaluated by the cumulative density and biomass of weeds, weed germination, viability of propagules and microsclerotia of soil pathogens such as *Pythium* sp., *Verticillium* sp., and *Fusarium* spp., percentage of diseased plants, as well as crop vigor and size [3, 4]. Traditionally, the crop vigor and plant health has been measured by crop diameter and leaf area. Recent agricultural studies have utilized image analyses for data collection [8–10]. Al-Tamimi et al. [11] estimated the area and biomass of rice leaves by optical imaging with RGB cameras. To obtain accurate image data for the crop size, however, the cameras should be in a fixed position, and it is challenging to do this in agricultural fields. Cockerton et al. [8] calculated the ratio of diseased area to healthy area of strawberry leaves by estimating green pixels and total pixels using images taken by an unmanned aerial vehicle. As an alternative to measuring the crop size; the leaf color can be used to measure crop health using chlorophyll reflectance and nutrient status. A crop can be unhealthy due to both biotic and abiotic stresses, and the leaf color can be changed as a result of metabolic and physiological changes [12]. If the steam application improves the health conditions of strawberry by controlling soilborne pests, it would be revealed by the leaf color.

Our hypothesis is that steam controls soil pests as well as chloropicrin fumigant and that the health of the crop can be detected by image analysis. The objective of this study is to determine if the image analysis of strawberry plant color spectrum, especially green, can be used to measure plant health during the crop establishment phase, and to estimate the relationship between plant color spectrum and fruit yield. In this study, the steam treatment is compared to the non-treated control and chloropicrin during two harvest seasons.

## Materials and methods

### Treatment description and data collection

Steam was applied on September 6, 2019 and September 11 and 12, 2020, respectively. This experiment was conducted at the Spence research farm 10 km south of Salinas, California, in a sandy loam soil with <1% organic matter. Each of the three treatments, nontreated control, steam, and chloropicrin, was replicated four times and arranged in a randomized complete block design.

In the 2019–2020 trial, the prototype field-scale steam applicator (Southern Turf Nurseries, STN, Elberta, AL) was equipped with a 223.7 kW h$^{-1}$ diesel-fueled Cleaver Brooks steam generator mounted on a trailer towed by a tractor (Fig 1). Steam was injected using a 3 m wide reverse tiller that was set to till 30 to 40 cm deep. Chloropicrin was applied at 24 mL m$^{-2}$ by subsurface drip irrigation. Strawberry plants (*Fragaria* × *ananassa* cv. 'Cabrillo') were transplanted on November 15, 2019. The strawberry fruit was harvested twice weekly for a total of 36 times from April to September 2020.

In the 2020–2021 trial, the self-propelled diesel fueled steam generator and applicator (JSE, Daegu, South Korea) was equipped with 20 straight shanks that could treat 2 m per pass (Fig 1). Steam was injected to a depth of 30 cm through a pipe on the back of the shanks. Chloropicrin 56.7% plus 1,3-Dichloropropene 37.1% mixture (PicClor 60 EC) was applied on October 24, 2020 at 26.19 mL m$^{-2}$. Strawberry plants (*Fragaria* × *ananassa* cv. 'Cabrillo') were transplanted on November 17, 2020. The strawberry fruit was harvested twice weekly for a total of 41 times from March to August 2021.

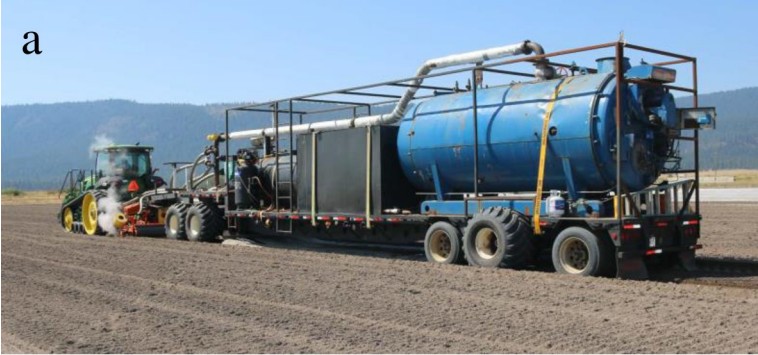

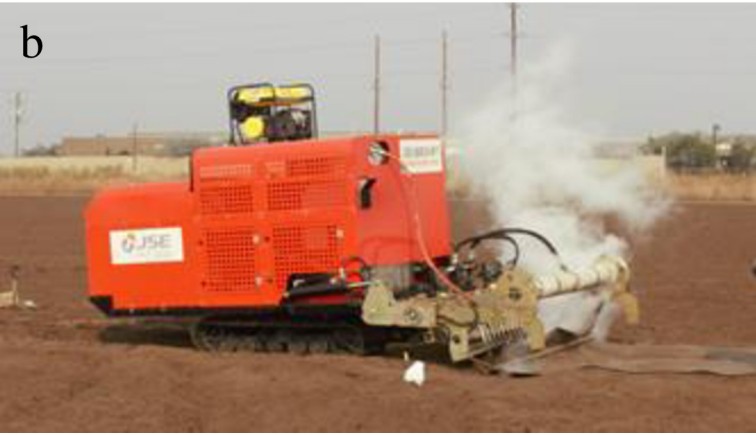

**Fig 1.** The steam generator and applicator used in 2019–2020 (a) and 2020–2021 (b) trials.

The soil temperature was measured by HOBO loggers (U12 Outdoor, Onset Computer Corp., Pocasset, MA) and the times above 65°C during the first hour after steam application at the depths of 10, 20, and 25 cm were 49.5, 57.0, and 23.5 minutes in the 2019–2020 trial and were 60.0, 60.0, and 0.0 minutes in the 2020–2021 trial, respectively.

Eight months after the treatments were applied, 6 m of each bed was photographed by a digital camera (EOS 70D DSLR, Cannon, Inc., Tokyo, Japan). The pictures of the three treatments are present in Fig 2. Each picture was converted from a JPEG file to a GIF image file for image analysis. Each GIF file was uploaded to the image analysis program available at http://mkwak.org/imgarea. The image analysis program outputs RGB codes and the number of pixels.

## Statistical methods

The image analysis program produced RGB codes with pixels occupied by each RGB code. There are various formulas to quantify colors using RGB codes, and there is no single formula to define "green" [13]. Instead of applying complicated formulas and restrictions, we observed all images and the associated RGB codes. There is a clear pattern that green colors satisfy both $G > R$ and $G > B$. In addition, green colors tend to be darker when $(R + B) / G$ is close to zero. For example, $(R, G, B) = (0, 43, 0)$ was relatively dark green, and $(R, G, B) = (102, 128, 102)$ was relatively light green. In addition, we identified the most five common RGB codes associated green colors: $(0, 43, 0)$, $(51, 85, 51)$, $(51, 85, 0)$, $(102, 128, 51)$, and $(102, 128, 102)$ ranging from dark green to light green. All other colors other than these green colors were removed to

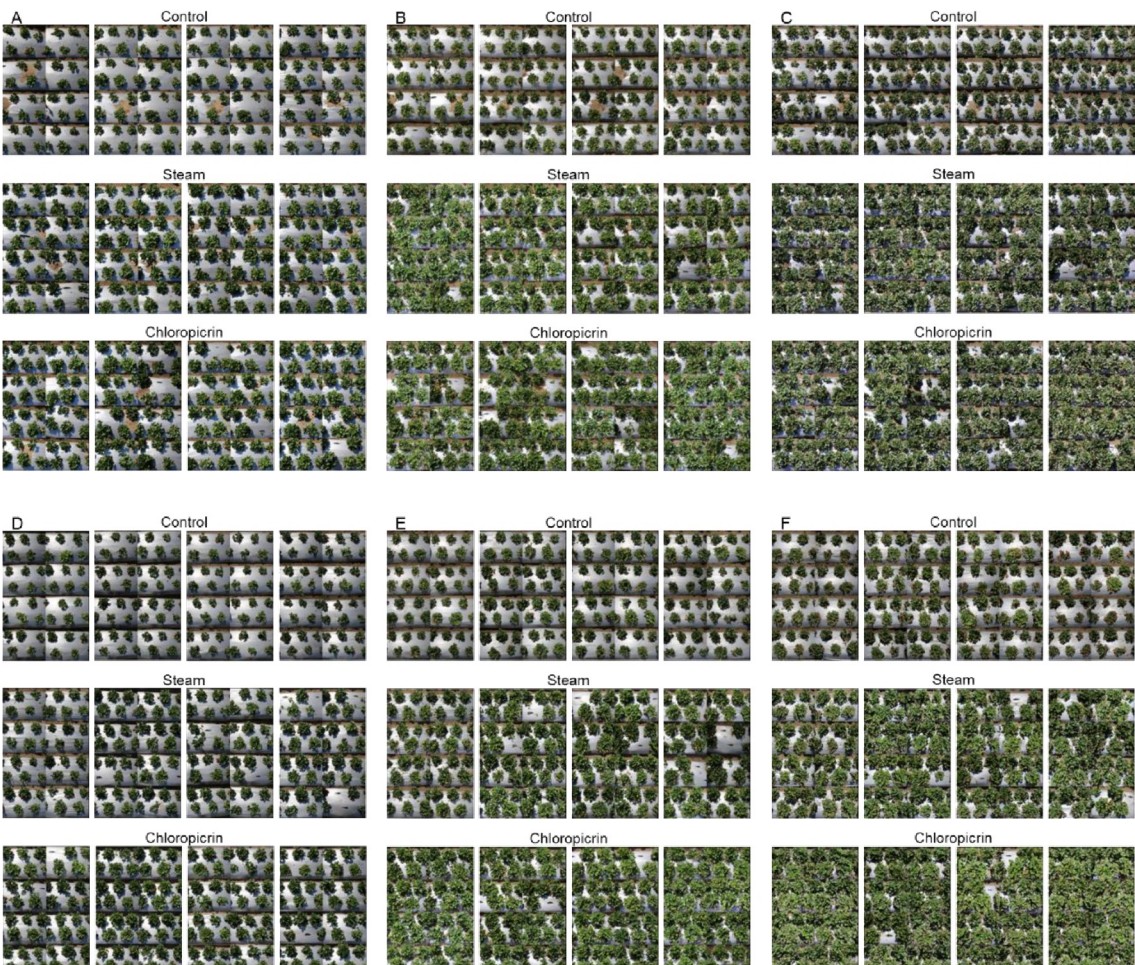

**Fig 2. Pictures for the image analysis.** A: May 2020; B: June 2020; C: July 2020; D: May 2021; E: June 2021; F: July 2021.

analyze green colors. We let $A_i$ denote the pixels occupied by the i-th green RGB code ($R_i$, $G_i$, $B_i$) for i = 1, 2, . . ., m, where m is the number of unique green colors identified by the image analysis program. To account for the proportion occupied by each color, we let $W_i = A_i / (A_1 + . . . + A_m)$ be the proportion, and the averages of R, G, B, R/G, and B/G were calculated by the weighted average $\Sigma W_i R_i$, $\Sigma W_i G_i$, $\Sigma W_i B_i$, $\Sigma W_i R_i /G_i$ and, $\Sigma W_i B_i /G_i$, respectively. Note that the weight $W_i$ quantifies the proportion occupied by each green color among the five common green colors (not all colors in the image). To compare the three treatments, the sampling distributions of these five statistics were approximated by the bootstrapping method [9, 14, 15]. To respect the varying pixels, the i-th green RGB code was sampled with a probability of $W_i$ in a bootstrap sample. Using 2,000 bootstrap samples, 95% bootstrap confidence intervals (CIs) were calculated for the difference of means; comparing chloropicrin to the nontreated control, steam to nontreated control, and steam to chloropicrin.

After visualizing the bootstrap distributions of the color variables, we formally compared the ratio of the area occupied by the dark green (0, 43, 0) to the area occupied by the light green (102, 128, 102). If the ratio value is above one, it indicates darker green strawberry leaves (healthier condition). The expected ratio was compared among the three treatments with respect to time (May, June, and July) at the significance level of α = 0.05. To account for the

repeated observations per experimental unit, the mixed-effects model was used for the hypothesis testing. To respect the normality assumption, the log-transformation was applied to the ratio. Finally, we compared the expected cumulative total fruit yield (grams per plant) among the three treatments at $\alpha = 0.05$. All statistical analyses were performed in R version 4.0.3 with the lmer and lmerTest packages [16–18].

## Results

Bootstrap sampling distributions of the five statistics (means of R, G, B, R/G, and B/G) are presented in Fig 3. Overall, the distributions of green colors, quantified by the RGB codes, appear different when the treatments of steam and chloropicrin were compared to the control, and the steam and chloropicrin treatments appear similar in both trials (2019–2020 and 2020–2021) as shown in Table 1. Note that green colors tend to be darker when R/G and B/G are close to zero, and Fig 3 demonstrated that the averages of R/G and B/G are generally lower in the steam treatment and chloropicrin treatment when compared to the control. Though the averages are not exactly the same, the order (the lowest in the chloropicrin treatment and the highest in the control) is consistent between the two seasons.

Various colors were sorted by the image analysis program (Fig 4), and the five common green colors were identified by the pixels occupied. The five RGB codes were (0, 43, 0), (51, 85, 51), (51, 85, 0), (102, 128, 51), and (102, 128, 102) in the order from dark to light. The

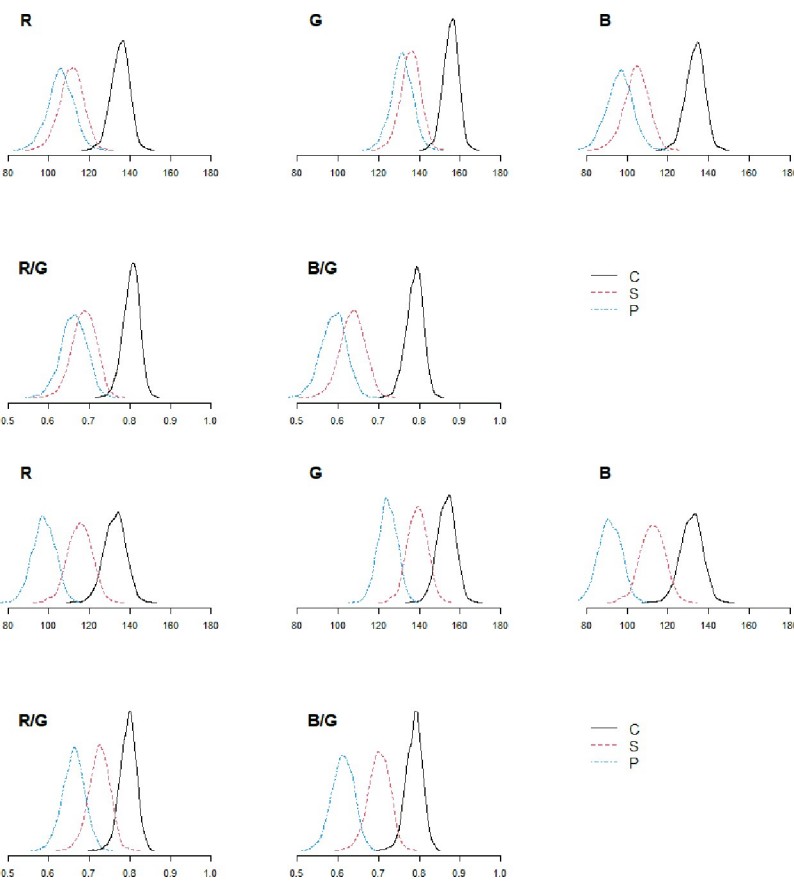

**Fig 3. Bootstrap sampling distributions for the means of R, G, B, R/G, and B/G values.** C, Non-treated control; S, Steam; P, Chloropicrin. The top panels were observed in the 2019–2020 trial, and the bottom panels were observed in the 2020–2021 trial.

**Table 1. Estimated averages of R, G, B, R/G, and B/G values (95% bootstrap CIs).**

| | Year | Control (C) | Chloropicrin (P) | Steam (S) | Difference (P − C) | Difference (S − C) | Difference (S − P) |
|---|---|---|---|---|---|---|---|
| R | 2020 | 135 (126, 143) | 106 (94, 117) | 111 (99, 122) | -29 (-44, -15) | -24 (-38, -10) | 5 (-12, 22) |
| | 2021 | 133 (122, 142) | 98 (86, 108) | 115 (104, 126) | -35 (-50, -20) | -17 (-32, -2) | 18 (2, 33) |
| G | 2020 | 155 (148, 162) | 131 (121, 141) | 136 (126, 145) | -24 (-36, -12) | -20 (-32, -8) | 4 (-10, 18) |
| | 2021 | 153 (144, 162) | 124 (115, 133) | 139 (129, 148) | -29 (-42, -17) | -14 (-27, -1) | 15 (1, 27) |
| B | 2020 | 133 (124, 142) | 96 (84, 108) | 104 (92, 116) | -37 (-52, -22) | -29 (-44, -14) | 8 (-9, 26) |
| | 2021 | 132 (121, 141) | 91 (80, 102) | 112 (100, 123) | -40 (-56, -25) | -19 (-34, -3) | 21 (5, 36) |
| R/G | 2020 | 0.80 (0.76, 0.84) | 0.66 (0.60, 0.72) | 0.69 (0.63, 0.74) | -0.14 (-0.21, -0.07) | -0.12 (-0.19, -0.04) | 0.02 (-0.06, 0.11) |
| | 2021 | 0.80 (0.76, 0.83) | 0.66 (0.61, 0.71) | 0.72 (0.67, 0.77) | -0.13 (-0.20, -0.07) | -0.07 (-0.13, -0.01) | 0.06 (-0.01, 0.13) |
| B/G | 2020 | 0.79 (0.74, 0.83) | 0.59 (0.53, 0.65) | 0.63 (0.57, 0.70) | -0.20 (-0.27, -0.12) | -0.15 (-0.23, -0.08) | 0.04 (-0.04, 0.13) |
| | 2021 | 0.79 (0.75, 0.82) | 0.61 (0.56, 0.66) | 0.70 (0.65, 0.75) | -0.18 (-0.24, -0.11) | -0.09 (-0.15, -0.02) | 0.09 (0.01, 0.16) |

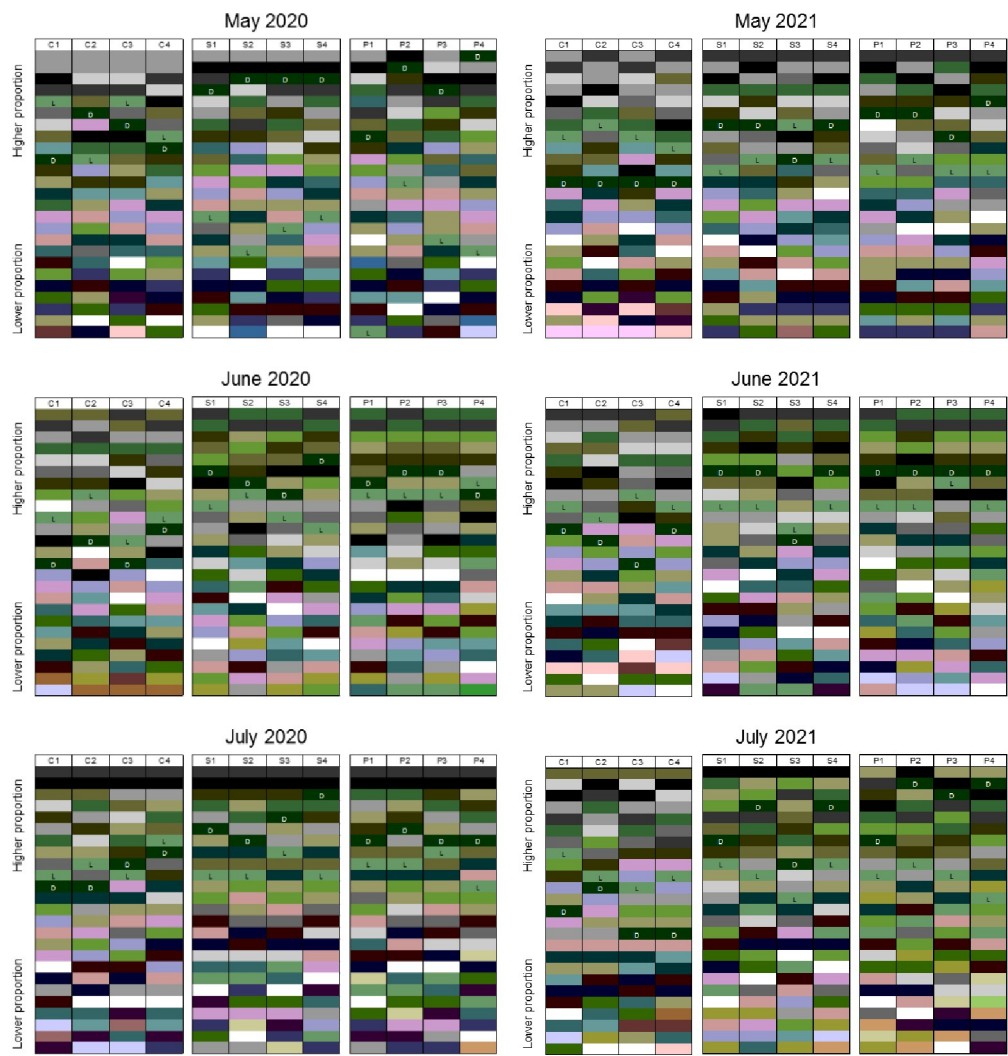

**Fig 4. The top 25 colors analyzed with Fig 2.** L, relatively light green ($R, G, B = 102, 128, 102$); D, relatively dark green ($R, G, B = 0, 43, 0$). C, Non-treated control; S, Steam; P, Chloropicrin.

percentage (%) covered by (0, 43, 0) and (102, 128, 102) were significantly different when the steam and chloropicrin treatments were compared to the nontreated control, the steam and chloropicrin treatments were similar (Fig 5). In both steam and chloropicrin treatments, we observed more dark green (0, 43, 0) than light green (102, 128, 102) in both trials (Fig 6), and we quantified the trend using the ratio of dark green area to light green area. Under the mixed-effects model, the log-transformed ratio at each month (from May to July) was substantially different when the steam treatment was compared to the nontreated control ($p < 0.0001$ in both 2020 and 2021 data) and when the chloropicrin treatment was compared to the control ($p < 0.0001$ in both 2020 and 2021 data). The difference between the steam and chloropicrin treatments was relatively small but was significant in the 2020–2021 trial ($p = 0.980$ in the 2020 data and $p = 0.041$ in the 2021 data). When normality was checked using the Shapiro test, and there was no strong indication of violation after using the log-transformed ratio. the p-values were 0.23 and 0.42 for the 2020 data and the 2021 data, respectively.

The trend of the ratio shown in Fig 6 was similar to the trend of the fruit yield (Fig 7). In the 2019–2020 trial, the estimated cumulative yield was 1,005 g/plant on average in the control group, and the average was 581 g/plant and 598 g/plant higher in the steam and chloropicrin treatments, respectively, compared to the nontreated control. In the 2020–2021 trial, it was 1264 g/plant on average in the control group, and it was 794 g/plant and 1095 g/plant higher in the steam and chloropicrin treatments, respectively.

Fig 8 displays that the ratio (dark green area to light green area) is related with the total fruit yield in both the 2019–2020 l and 2020–2021 trials, and estimated coefficient of determination were $R^2 = 0.78$ and 0.46, respectively. The coefficient of determination in 2020–2021 trial was lower than that in 2019–2020 trial but the estimated slope was steeper.

## Discussion

The both trials, the difference of the soil temperature in the steam treatments were caused by the difference of the steam machines (Fig 1). As described in the Materials and Methods section, the steam heat penetrated deeper in 2019–2020 trial than in 2020–2021 trial. But the

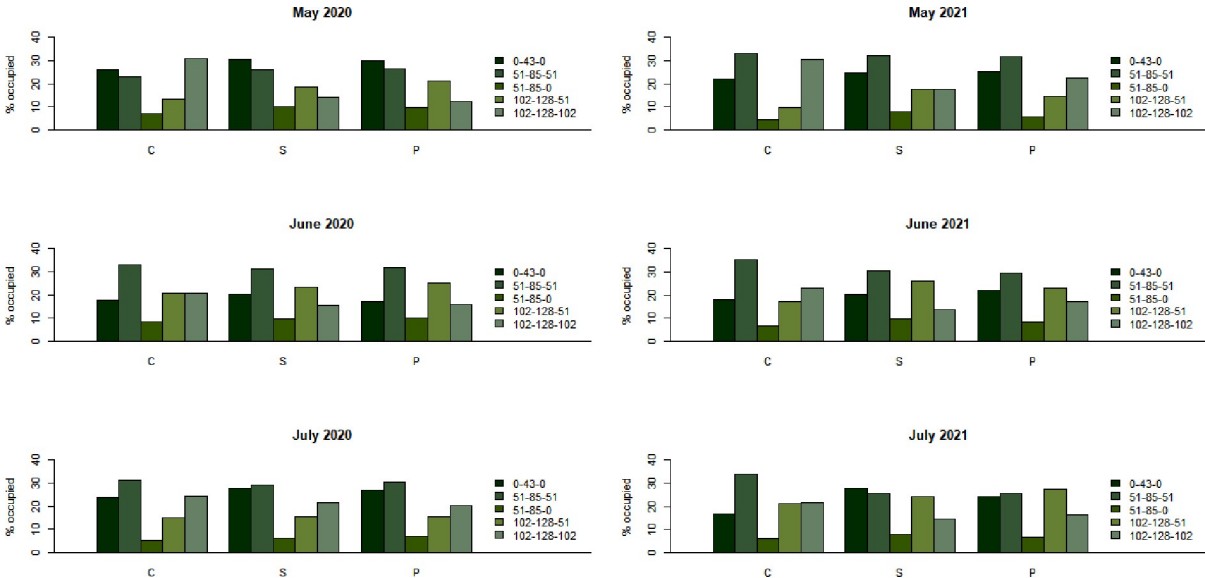

**Fig 5. The percent occupied by the five most common green RGB codes.** C, Non-treated control; S, Steam; P, Chloropicrin.

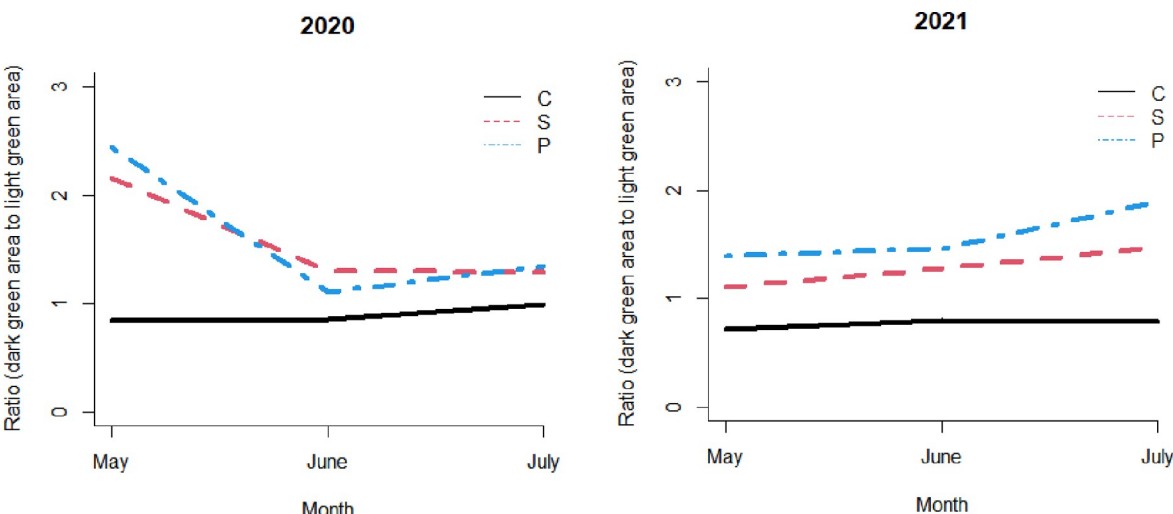

**Fig 6. The color ratio of the dark green (R, G, B = 0, 43, 0) to the light green (R, G, B = 102, 128, 102).** C, Non-treated control; S, Steam; P, Chloropicrin.

steam heat persisted longer in the 2020–2021 than in 2019–2020 trial. This difference of the soil temperature may be the cause of the different results between the two trials, and there are many previous studies proving this.

Soil borne pests are important factors that harm the health of crops. In the strawberry industry, *Pythium* and *Verticillium* spp. are important soilborne diseases. *Pythium* spp. causes black root rot and can reduce fruit yield while *Verticillium* spp. causes vascular deterioration and wilting that results in crop death, [19, 20]. Weeds are a major limitation in efficient crop production as they result in yield loss and quality reductions due to competition between crops and weeds and hand removal of weeds is expensive. In the trials, burclover (*Medicago polymorpha*), burning nettle (*Urtica urens*), chickweed (*Stellaria media*), common groundsel (*Senecio vulgaris*), henbit (*Lamium amplexicaule*), lesser swinecress (*Lepidium didymum*), and perennial ryegrass (*Lolium perenne*) are common weeds [21]. Steam has been applied for

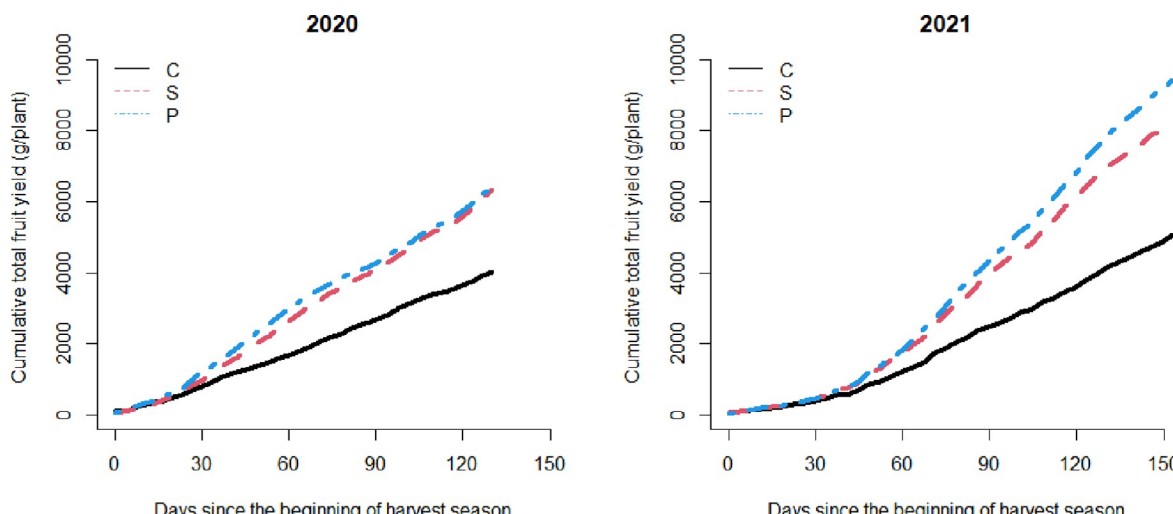

**Fig 7. The cumulative total fruit yield during two harvest seasons.** C, Non-treated control; S, Steam; P, Chloropicrin.

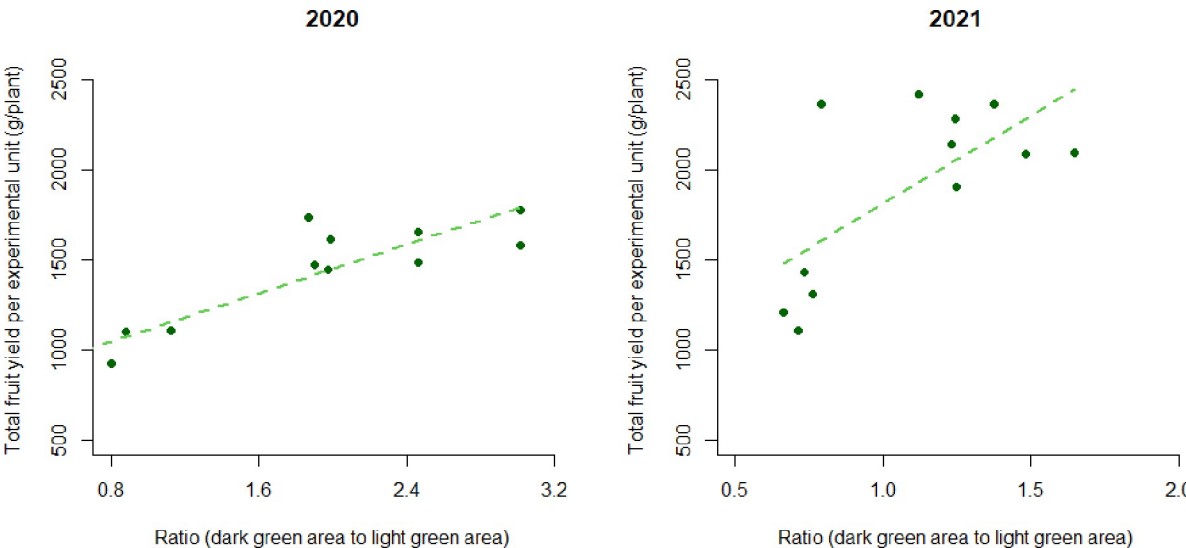

**Fig 8. The total fruit yield versus the ratio of dark green area to light green area observed in May per experimental unit (4 control, 4 steam, and 4 chloropicrin for both years 2020 and 2021; $r^2$ = 0.78 and 0.46, respectively).**

control of pests such as the phytopathogens and weeds in strawberry fields as an environment friendly alternative to MB [4–6]. To reduce important strawberry pathogens and weed seedling emergence, soil must be treated with steam at 60°C for 3 minutes [22, 23]. The soil temperature observed in this experiment was maintained at 65°C for more than 20 minutes to a depth of 25 cm, which was enough to reduce phytopathogens and weed seed viability as demonstrated in previous studies [3, 4, 24]. Production of strawberry fruit in soils disinfested with steam was higher than that of the non-treated control [4, 6, 7]. Our results indicated that the total fruit yield in the steam treatment was similar to that in the chloropicrin treatment but also the yield was highly correlated with the ratio of dark green area to light green area.

The use of digital image analysis has been increasing dramatically over the last several years in the plant sciences such as agriculture, horticulture, pathology, forestry, and ecology, as software has become more sophisticated and user-friendly [25]. It is not only for convenience but also a sensitive method of measuring plant vigor in a timely manner. Therefore, digital data collection is more efficient than traditional methods. Plant growth can be rapid. Increased efficiency means we are less likely to miss a key time when there are real differences between treatments. Kim et al. [9] applied the bootstrapping method to analyze RGB codes for comparing weed densities instead of manual counts. In that study, the focus was the proportion of green area but here our focus was the degree of greenness (relatively dark or light). Image analysis to measure plant size is subject to error due to need for a fixed distance from the target. Our method avoided this problem by focusing on the degree of greenness which did not require a fixed distance from the leaf to determine plant health.

Image analysis techniques are capable of detecting small differences in the distribution of various colors among the images which is challenging for humans to perform. For instance, in case of strawberry plants, there are two major colors, green in the chlorophylls and red in the anthocyanins. The green and red colors in strawberry plants are mainly found in leaves and fruits, respectively. We decided to analyze only green color to evaluate plant health because the green color of strawberry plant leaves reflects the chlorophyll reflectance and nutrient status [26]. Furthermore, leaf greenness (SPAD value) and root fresh weight of strawberry plants were positively correlated to chlorophyll reflectance [27]. Where there are biotic and abiotic

stresses to plants, the leaves of the plants will be light green. Our image analysis results suggest that there is a relationship between two specific green colors and the health and yield of strawberry plants (Table 1). Sun et al. [28] reported that rice leaves become greener and larger with a greater nitrogen supplement. So, if the strawberry roots and vascular system are growing in soils with few pathogens then they can more readily uptake nutrients like nitrogen and would be darker green. Moreover, no difference of the total pixel percentage of the dark and light green colors in the steam and chloropicrin treatments might mean that the steam application was as effective as the chloropicrin application for soil disinfestation as the chloropicrin application. Fig 5 also showed that the dark green (R, G, B = 0, 43, 0) is more abundant than the light green (R, G, B = 102, 128, 102) in the steam and chloropicrin treatments, and the reverse is shown in the control. Hence, the R/G and B/G values of the steam and chloropicrin treatments were close to zero, compared to one in the nontreated control. For instance, the R/G and B/G values of the dark green (R, G, B = 0, 43, 0) is 0 and the values of the light green (R, G, B = 102, 128, 102) are 0.80.

Regarding the steam application of bootstrapping to the image analysis, Kim et al. [9] found that the bootstrap method applied to RGB codes results in higher statistical power than the Dunnett-Tukey-Kramer test which is a traditional multiple comparison test. If such a traditional statistical test was applied to this study, it would treat the entire data as a sample of size four per treatment which would be too scarce to compare the two treatments to the control. Instead, we decided to utilize the RGB codes and the pixels to increase statistical information for the comparisons (Fig 5). Consequently, the method well demonstrated the efficacy of steam and chloropicrin in our study.

## Conclusion

Steam application improved the growth and productivity of strawberry plants as much as the commercial fumigant chloropicrin. The image analysis with the appropriate statistical strategy was very useful for the comparison of the plant growth among the treatments, and it was easier and faster than traditional manual methods such as counting and weighing. Taking pictures in large fields, however, is still a time-consuming process. In a future study, an unmanned aerial vehicle will overcome the posed challenge, and we will need to test for the reliability. Advancements in technological data collection, image analysis method, and statistical strategy will increase the accuracy and efficiency of plant growth measurement and fruit yield prediction.

## Supporting information

**S1 Data.**
(PDF)

## Author Contributions

**Conceptualization:** Dong Sub Kim.

**Data curation:** Steven B. Kim.

**Formal analysis:** Steven B. Kim, Dong Sub Kim.

**Funding acquisition:** Steven A. Fennimore.

**Investigation:** Steven A. Fennimore.

**Methodology:** Steven B. Kim, Dong Sub Kim.

**Project administration:** Steven A. Fennimore.

**Resources:** Steven A. Fennimore.

**Software:** Steven B. Kim.

**Supervision:** Steven A. Fennimore.

**Validation:** Steven B. Kim, Steven A. Fennimore, Dong Sub Kim.

**Visualization:** Steven B. Kim, Dong Sub Kim.

**Writing – original draft:** Steven B. Kim, Steven A. Fennimore, Dong Sub Kim.

**Writing – review & editing:** Steven B. Kim, Steven A. Fennimore, Dong Sub Kim.

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
