## [Decision Letter · Decision Letter 0]

19 May 2022

PONE-D-22-01998A potential relationship between soil disinfestation efficacy and leaf green reflectancePLOS ONE

Dear Dr. Kim,

Thank you for submitting your manuscript to PLOS ONE. Firstly, we would like to apologize for the delay in processing your manuscript. It has been exceptionally difficult to secure reviewers to evaluate your study. We have now received one completed review, which is available below. The reviewer has raised significant scientific concerns about the study that need to be addressed in a revision.

Please note that we have only been able to secure a single reviewer to assess your manuscript. We are issuing a decision on your manuscript at this point to prevent further delays in the evaluation of your manuscript. Please be aware that the editor who handles your revised manuscript might find it necessary to invite additional reviewers to assess this work once the revised manuscript is submitted. However, we will aim to proceed on the basis of this single review if possible.

After careful consideration, we feel that it has merit but does not fully meet PLOS ONE’s publication criteria as it currently stands. Therefore, we invite you to submit a revised version of the manuscript that addresses the points raised during the review process.

We look forward to receiving your revised manuscript.

Kind regards,

Miquel Vall-llosera Camps

Senior Editor

PLOS ONE

Journal Requirements:

Reviewers' comments:

Reviewer's Responses to Questions

**Comments to the Author**

1. Is the manuscript technically sound, and do the data support the conclusions?

Reviewer #1: Partly

2. Has the statistical analysis been performed appropriately and rigorously? 

Reviewer #1: Yes

3. Have the authors made all data underlying the findings in their manuscript fully available?

Reviewer #1: No

4. Is the manuscript presented in an intelligible fashion and written in standard English?

Reviewer #1: Yes

5. Review Comments to the Author

Reviewer #1: The present manuscript deals with the soil disinfestation efficacy assessment using color digital photography. The manuscript is readable and some interesting results were obtained. However, I have some concerns like

The treatments applied for the year 2019-2020 and 2020-2021 were significantly different. Under such a situation, can these be compared? This is clearly visible from Fig. 3. In year 2019-2020, Chloropicrin and steam treatments were almost similar while it was different in 2020-2021.

The classification of RGB codes in the order from dark to light green seems quite arbitrary. It should be concrete based on solid statistics or physics principle.

My specific comments are

L27: “We hypothesized that plant health could be quantified and treatments differentiated” to “We hypothesized that plant health could be quantified and treatments can be differentiated”

L64-66: Delete “Nitrogen and magnesium deficiencies in strawberry plants results in low chlorophyll concentration in leaves [12]”

L82: Provide the full form of “OM”

L87: Was “Chloropicrin” sprayed on bare soil?

L98: “(PicClor 60 EC) was applied October 24, 2020” to “(PicClor 60 EC) was applied on October 24, 2020”

L110: What do you mean by RGB codes and area occupied by RGB code? Please describe in detail.

L125: Was the weight calculated for each color or only for green?

L141: Which packages were used in R software? Test the normality of the ratio and log transformed ratio.

L192: “correlation coefficients” to “coefficient of determination”

L193: “correlation coefficients” to “coefficient of determination”

L202: “Soil pests” to “Soil borne pests”

L206-207: “weeds and hand removal is expensive.” To “weeds. Hand removal of weeds is expensive.”

L211: “environmentally friendly” to “environment friendly”

L226-227: Sentence is incomplete

L238-242: Meaningless, please rewrite.

L247-248: “effective of a method of soil disinfestation as the chloropicrin application.” To “effective as the chloropicrin application for soil disinfestation.”

L250: “treasures” to “treatments”

L253: For which treatment?

6. PLOS authors have the option to publish the peer review history of their article (what does this mean?). If published, this will include your full peer review and any attached files.

Reviewer #1: **Yes: **Bappa Das

---

## [Author Response · Author response to Decision Letter 0]

10 Jun 2022

The treatments applied for the year 2019-2020 and 2020-2021 were significantly different. Under such a situation, can these be compared? This is clearly visible from Fig. 3. In year 2019-2020, Chloropicrin and steam treatments were almost similar while it was different in 2020-2021.

Response

Thank you for your comment. Though the treatment applications were not identical, we observed similar trends between the two seasons. In Fig 3., the locations (centers) of the distributions have the same order. In addition, Figs 2, 5, 6, and 7 show similar trends between S and P which are distinguishable from C (The yields both years followed the same trends.). Therefore, the overall effect of S and C on the plant health (observed by the colors) and the productivity are similar. Here is another important point that we needed to include in the discussion. We are aware that the both trials, the difference of the soil temperature in the steam treatments were caused by the difference of the steam machines (Fig. 1). This difference of the soil temperature may be the cause of the different results between the two trials, and there are many previous studies proving this. This inserted into the discussion (Line 215).

The classification of RGB codes in the order from dark to light green seems quite arbitrary. It should be concrete based on solid statistics or physics principle.

Response

Our choice was based on statistics (i.e., calculated from the data). In general, based upon our knowledge, there are no definite classifications of “dark green” and “light green,” and there are a number of (complicated) ways to combine R, G, and B for color analysis (e.g., Angulo & Serra, 2007). Instead of using the complicated formulas, which were not helpful in our study, we matched observed images and distributions of RGB codes. While green colors have the common property of G > R and G > B, a green color tends to be darker when (R + B) / G is close to zero (i.e., R/G and B/G are close to zero). It is statically shown in Fig 3, and the observed pattern (the order) is consistent between the two seasons. We added this explanation in the section of statistical method (Line 118) and in the section of results (Line 155).

L110: What do you mean by RGB codes and area occupied by RGB code? Please describe in detail.

Response

Line 109 ‘The image analysis program outputs RGB codes per image file and areas (pixels) occupied by each RGB code.’ to ‘The image analysis program outputs RGB codes and the number of pixels.’

L126: Was the weight calculated for each color or only for green?

Response

The weights were only for green, and we added one sentence (Line 130). Thank you for helping us making the manuscript clearer.

L141: Which packages were used in R software? Test the normality of the ratio and log transformed ratio.

Response

lmer and lmerTest packages were used for the mixed-effects model (Bates et al., 2015; Kuznetsova et al., 2017), and we added the related references. Regarding the normality, we tested the normality of the log-transformed ratio because it is what we used in the analysis. We did not observe severe violations of normality. The p-values calculated from the Shapiro test were 0.23 and 0.42 for the 2020 data and the 2021 data, respectively, and we reported in the results section (Line 146). Thank you.

L27: “We hypothesized that plant health could be quantified and treatments differentiated” to “We hypothesized that plant health could be quantified and treatments can be differentiated”

L64-66: Delete “Nitrogen and magnesium deficiencies in strawberry plants results in low chlorophyll concentration in leaves [12]”

L82: Provide the full form of “OM”

L87: Was “Chloropicrin” sprayed on bare soil?

L98: “(PicClor 60 EC) was applied October 24, 2020” to “(PicClor 60 EC) was applied on October 24, 2020”

L192: “correlation coefficients” to “coefficient of determination”

L193: “correlation coefficients” to “coefficient of determination”

L202: “Soil pests” to “Soil borne pests”

L206-207: “weeds and hand removal is expensive.” To “weeds. Hand removal of weeds is expensive.”

L211: “environmentally friendly” to “environment friendly”

L226-227: Sentence is incomplete

L238-242: Meaningless, please rewrite.

L247-248: “effective of a method of soil disinfestation as the chloropicrin application.” To “effective as the chloropicrin application for soil disinfestation.”

L250: “treasures” to “treatments”

L253: For which treatment?

Response

Revised as suggested.

---

## [Decision Letter · Decision Letter 1]

6 Jul 2022

A potential relationship between soil disinfestation efficacy and leaf green reflectance

PONE-D-22-01998R1

Dear Dr. Kim,

We’re pleased to inform you that your manuscript has been judged scientifically suitable for publication and will be formally accepted for publication once it meets all outstanding technical requirements.

Kind regards,

Aradhana Mishra, Ph.D.

Academic Editor

PLOS ONE

Additional Editor Comments (optional):

Reviewers' comments:

Reviewer's Responses to Questions

**Comments to the Author**

1. If the authors have adequately addressed your comments raised in a previous round of review and you feel that this manuscript is now acceptable for publication, you may indicate that here to bypass the “Comments to the Author” section, enter your conflict of interest statement in the “Confidential to Editor” section, and submit your "Accept" recommendation.

Reviewer #1: All comments have been addressed

Reviewer #2: All comments have been addressed

2. Is the manuscript technically sound, and do the data support the conclusions?

Reviewer #1: Yes

Reviewer #2: Yes

3. Has the statistical analysis been performed appropriately and rigorously? 

Reviewer #1: Yes

Reviewer #2: Yes

4. Have the authors made all data underlying the findings in their manuscript fully available?

Reviewer #1: Yes

Reviewer #2: Yes

5. Is the manuscript presented in an intelligible fashion and written in standard English?

Reviewer #1: Yes

Reviewer #2: Yes

6. Review Comments to the Author

Reviewer #1: Authors have addressed all the comments satisfactorily and I can suggest the publication of the manuscript in Plos One.

Reviewer #2: In the manuscript entitled "A potential relationship between soil disinfestation efficacy and leaf green reflectance", the authors have evaluated steam as an alternative method of soil disinfestation to fumigation. In my opinion, the manuscript is sound and has the potential to be published.

Minor comment

Please include the full form of any abbreviations used in the abstract of this manuscript.

7. PLOS authors have the option to publish the peer review history of their article (what does this mean?). If published, this will include your full peer review and any attached files.

Reviewer #1: **Yes: **Bappa Das

Reviewer #2: **Yes: **Sahil Mahfooz

---

## [Editor Report · Acceptance letter]

14 Jul 2022

PONE-D-22-01998R1 

A potential relationship between soil disinfestation efficacy and leaf green reflectance 

Dear Dr. Kim:

I'm pleased to inform you that your manuscript has been deemed suitable for publication in PLOS ONE. Congratulations! Your manuscript is now with our production department. 

Kind regards, 

on behalf of

Dr. Aradhana Mishra 

Academic Editor

PLOS ONE